**Understanding representations of uncertainty, an eye-tracking study part II: The effect of expertise**

Louis Williams[1,5], Kelsey J. Mulder[2,3], Andrew Charlton-Perez[2], Matthew Lickiss[4], Alison Black[4], Rachel McCloy[5], Eugene McSorley[5], Joe Young[6]

[1]ICMA Centre, Henley Business School, University of Reading, Whiteknights, PO Box 242, Reading, RG6 6BA, United Kingdom.

[2]Department of Meteorology, Earley Gate, University of Reading, Whiteknights Road, PO Box 243, Reading, RG6 6BB, United Kingdom.

[3]Liberty Specialty Markets, 20 Fenchurch Street, London EC3M 3AW, UK

[4]Department of Typography & Graphic Communication, School of Arts, English and Communication Design, No. 2 Earley Gate, University of Reading, Whiteknights Road, PO Box 239, Reading RG6 6AU.

[5]School of Psychology and Clinical Language Sciences, Earley Gate, University of Reading, Whiteknights Road, PO Box 238, Reading, RG6 6AL, United Kingdom.

[6]Department of Atmospheric Sciences, University of Utah, 115, Salt Lake City, UT 84112, United States

Correspondence to: Louis Williams (louiswilliams@dynamicplanner.com)

**Abstract.** As the ability to make predictions of uncertainty information representing natural hazards increases, an important question for those designing and communicating hazard forecasts is how visualisations of uncertainty influence understanding amongst the intended, potentially varied, target audiences. End-users have a wide range of differing expertise and backgrounds, possibly influencing the decision-making process they undertake for a given forecast presentation. Our previous, linked study (Mulder et al, 2023), examined how the presentation of uncertainty information influenced end-user decision making. Here, we shift the focus to examine the decisions and reactions of participants with differing expertise (Meteorology, Psychology and Graphic Communication students) when presented with varied hypothetical forecast representations (boxplot, fan plot or spaghetti plot with and without median lines), using the same eye-tracking methods and experiments. Participants made decisions about a fictional scenario involving the choices between ships of different sizes in the face of varying ice thickness forecasts. Eye-movements to the graph area and key, and how they changed over time (early, intermediate, and later viewing periods), were examined. More fixations (maintained gaze on one location) and time fixating was spent on the graph and key during early and intermediate periods of viewing, particularly for boxplots and fan plots. The inclusion of median lines led to less fixations being made to all graph types during early and intermediate viewing periods. No difference in eye movement behaviour was found due to expertise, however those with greater expertise were more accurate in their decisions, particularly during more difficult scenarios. Where scientific producers seek to draw users to the central estimate, an anchoring line can significantly reduce cognitive load leading both experts and non-experts to make more rational decisions. When asking users to consider extreme scenarios or uncertainty, different prior expertise can lead to significantly different cognitive load for processing information with an impact on ability to make appropriate decisions.

## 1. Introduction

The importance of understanding the most ideal approach for communicating uncertainty information is a common across multiple domains in everyday life and across a range of sciences (Fischhoff, 2012) and is an established problem in geoscience communication (Stephens et al, 2012). This importance has been highlighted by the current COVID-19 pandemic during which there has been a sharp increase in the use of unfamiliar visualizations of uncertainty presented to the public in order to explain the basis of decisions made to justify the response being asked of them to adopt modified and new behaviours in order to mitigate transmission. As more unfamiliar and detailed information is presented to

and interpreted by non-specialists, the decisions made as a result have a significant impact on health, society and the environment, so careful consideration of communication is essential (Peters, 2008). It is clear that people have trouble gaining an appropriate understanding of uncertainty information and how best to use this in order to support optimal decisions (e.g., Tversky and Kahneman, 1974; Nadav-Greenberg and Joslyn, 2009; Roulston and Kaplan, 2009; Savelli and Joslyn, 2013). A great deal of research has been concerned with addressing the most appropriate way to communicate uncertainty to promote effective decision-making and understanding (Fischhoff, 2012; Milne et al., 2018). Deciding what uncertainty information should be included, what ought to be emphasized, and the manner in which it is best conveyed all have an important role to play (Bostrom et al., 2016; Broad et al, 2012; Morss et al., 2015; Padilla et al., 2015). Furthermore, there is a reluctance by authors, such as data scientists, journalists, designers and science communicators, to present visual representations of quantified uncertainty (Hullman 2019). There is a belief that it will overwhelm the audience and the main purpose of the data, invite criticism and scepticism, and that it may be erroneously interpreted as incompetence and a lack of confidence which will encourage a mistrust of the science (Fischhoff, 2012; Gistafson & Rice, 2019; Hullman, 2019). This research points to the lack of consistent recommendations and stresses the need for the form of communication being tailored to both the aims and desired outcomes of the communicator and the needs and abilities of the audience (Spiegelhalter et al., 2011; Lorenz et al., 2015; Harold et al., 2016; Petropoulos et al., 2022).

Visualizing uncertainty in geoscience forecasts needs to balance robustness, richness, and saliency (Stephens, et al. 2012). Recently, numerous examples of this have focussed on creative ways to achieve this (Lorenz et al., 2015; Harold et al., 2016; Petropoulos et al., 2022). Communication of uncertainty can take the forms of words, but this can lead to issues of ambiguity caused by the language used and the variation in user interpretation (Wallsten et al, 1986; Skubisz et al., 2009). However, there is clearly strength to this approach when it is needed. For example, taking a storyline approach has been shown to be a powerful technique for communicating risk when less focus is needed on probabilistic information and more emphasis is needed on plausible future events (Shepherd et al., 2018; Sillmann et al., 2021). To overcome issues of ambiguity of words, numbers are often used to present uncertainty as probabilities in the form of fractions (1/100), natural frequencies (1 in 100), or percentages (1%), but these forms can lead to ratio bias or denominator neglect (Morss et al., 2008; Kurz-Milcke et al., 2008; Reyna and Brainerd, 2008; Denes-Raj and Epstein, 1994; Garcia et al., 2010), and the most effective form to use to aid understanding can depend on the context (Gigerenzer & Hoffrage, 1995; Joslyn & Nichols, 2009). Similarly presenting uncertainty graphically can take many forms which means they have the advantage of

flexibility of presentation, can be tailored for specific audiences, can help with differing levels
of numeracy and can help people focus on the important gist of the information when using
uncertainty to help reach a decision (Feldman-Stewart et al., 2007; Peters et al, 2007; Lipkus
and Holland , 1999). As with the use of words, the choice of graphic to employ is dependent
on the audience and intended message outcome (Spiegelhalter, 2017) and can lead to the
overestimation of risk and negative consequences depending on the framing of the
information (Vischers et al, et al, 2009). Pie charts are good for presenting proportions and
part-to-whole comparisons and benefit from being intuitive and familiar to the public, but
interpretation can sometimes be difficult (Nelson et al., 2009). Bar charts are useful for
communicating magnitude and allowing comparisons (Lipkus, 2007) while line graphs are
helpful in conveying trend information about the change in uncertainty over time. Icons can
also be very useful, especially so for people with low numeracy and have been found to be
effective when supplemented by a tree diagram (Galesic et al., 2009; Gigerenzer et al, 2007;
Kurz-Milcke et al., 2008). These types of graphical communication can also include
information about the range of uncertainty (such as a "cone of uncertainty", Morss et al.,
107   2016).

Previous research has shown that including uncertainty information can aid users to make
more rational decisions (Nadav-Greenberg et al., 2008; Nadav-Greenberg and Joslyn, 2009;
Roulston and Kaplan, 2009; Savelli and Joslyn, 2013 St John et al., 2000). One way in which
this is achieved is by use of heuristics (Tversky and Kahneman, 1974). If selected wisely
then these can help simplify probabilistic information to bolster and speed decisions promote
optimal interpretation of data. However, poor selection can hinder and encourage suboptimal
decisions (Mulder et al., 2020). For example providing an anchor value alongside data can
help users interpret the data more efficiently by focussing them on that particular value (for
example, focussing people on precipitation level on days like this as a start point to
estimating rainfall) but if chosen poorly can encourage a more extreme and suboptimal
interpretation (focussing on the maximum precipitation level on days like this would
encourage higher estimates of rainfall). In terms of graphical visualization of uncertainty,
providing a central line showing a likely hurricane track has been reported to distract users
from possible hurricane tracks given by the cone of uncertainty. Equally, however, the cone
of uncertainty has been sometimes misinterpreted as showing the extent of the storm (Broad
et al., 2007). Beyond heuristics, other design choices have also been found to affect optimal
and efficient decision-making (Speier, 2006; Kelton et al., 2010; Wickens et al., 2021).
Different designs of boxplots and graphs showing the same information affect decisions and
interpretations (Correll and Gleicher, 2014; Bosetti et al., 2017; Tak et al., 2013, 2015).
Forecasting maximum values from graphs was found to depend on graph type (Mulder et al.,
2020). Giving tornado warnings with probabilistic information about where a tornado may
strike increased response in those areas compared with deterministic information (Ash et al.,

130    2014).

Part I of this study, which from here will be called "companion paper" (Mulder et al., 2023),
shows that, for all groups, great care is needed in designing graphical representations of
uncertain forecasts. This is especially so when attention needs to be given to critical
information, and the presentation of the data makes this more difficult. In particular, well
known anchoring effects associated with mean or median lines can draw attention away
from extreme values for particular presentation types (Broad et al., 2007; Nadav-Greenberg
et al. 2008; Mulder et al., 2020). The availability of easy-to-use tools that make the
development of complex graphical representations of forecasts quick and cheap to produce,
poses new challenges for the geo-scientists. Within the environmental sciences, making
forecasts of natural hazards (such as landfall of hurricanes, flooding, seismic risk and the
changing climate) useful to end-users depends critically on communicating in a concise and
informative way. Particularly as end-users have a wide range of differing expertise, spanning
a spectrum between geo-physical scientists to those with no formal scientific training.
Therefore, the way in which information is displayed is very important for avoiding
misperceptions and ensuring appropriate steps are taken by end-users, especially when
perceptions of natural hazards can differ between experts and non-experts (Fuchs et al.,
2009; Goldberg & Helfman, 2010). Here, we compare the response of three different groups
of end-users with different levels of scientific expertise to the same series of forecast
presentations to explore how more and less complex presentations influence decision
making and perception.
Expertise differences may be due to greater familiarity with the ways in which hazard
information is made available. This enables experts to make more economically rational
decisions and to interpret uncertainty information more effectively (Mulder et al., 2020).
However, the role of expertise remains unclear with some studies showing no differences in
decision-making tasks with both experts and non-experts able to process and use forecast
information to make decisions, with the inclusion of uncertainty information found to be
useful for both experts and non-experts (Nadav-Greenberg et al., 2008; Kirschenbaum et al.,
2014; Wu et al., 2014). Furthermore, it is unclear whether presentation of uncertainty
information in visual formats results in benefits over using verbal and numerical expressions.
For instance, uncertainty presented as pictograph or graphical representations may help with
understanding and interpretation (Zikmund-Fisher et al., 2008; Milne et al., 2015; Susac et
al., 2017). Additionally, research is required to examine differences in expertise, particularly
as deterministic construal errors can be made as observers are often unaware that
uncertainty is being depicted within visualisations (Joslyn & Savelli, 2021). Inappropriate
information that captures attention is also often relied on, which can distort judgements
(Fundel et al., 2019).
Experts are better at directing attention (through eye movements) to the important
information required for making a decision. For example, in judgments of flight failures,
expert pilots were found to make faster and more correct decisions, making more eye
movements to the cues related to failures than non-experts (Schriver et al, 2008). Kang and
Landry (2014) also found non-experts to improve after they were trained with the eye
movement scan paths of experts; training led non-experts to make fewer errors (false
alarms) on aircraft conflict detection tasks. However, there is little research examining eye
movements when experts and non-experts are required to make decisions using graphical
and numerical forecast information. It is not clear which aspects of forecast information are
being examined and when, and equally which, are being ignored.
More generally, research has shown that when viewing images, more fixations are made to
informative regions and areas of interest (Unema et al., 2005). The times at which these
fixations are made has been found to vary depending on task, decision type and expertise.
Antes (1974) found that early fixations, in the first few seconds of viewing pictures, were
towards informative areas. Goldberg and Helfman (2010) also showed that important regions
of interest were fixated early during observation of different graphs. Experts have been
shown to identify and fixate informative aspects of visual information more quickly and more
often than non-experts (Maturi & Sheridan 2020; Charness, Reingold, Pomplun, &
Stampe, 2001; Kundel, Nodine, Krupinski, & Mello-Thoms, 2008). As well as informative
parts of a scene or image, Shimojo et al. (2003) reported that the likelihood that fixation
would be made to the item preferred, increased over time, particularly in the final second
before selection (see also Glaholt & Reingold, 2009; Simion & Shimojo, 2006; Williams et al.,
2018). These results show that informative and preferred areas of images are selectively
fixated early on, more often and for longer. As viewing evolves, fixations start to reflect final
choices and preferences. The temporal development of this is task-dependent and
influenced by expertise.
Here, we explore eye movement behaviour to similar hypothetical scenarios but with
particular interest on differences due to participant expertise/background, following the
research discussed, of gaze to graph areas and keys over different time periods of the
decision-making process. Regardless of expertise, the presence of a median line on graphs
has been found to influence the location of participants gaze fixations moving their
distributions closer to the median line (Mulder et al, 2020). Depending on graph type the

presence of a key can lead to errors which may be function of finding that the key is not directly fixated in those representations (Mulder et al., 2020.  Here we explore these patterns, in particular whether these are a function of expertise. As in our companion paper (Mulder et al., 2023), we examine gaze patterns when faced with the task of making decisions about a fictional scenario involving the choices between ships of different sizes in the face of varying ice thickness forecasts (30%,50%,70%), when presented in different formats (boxplot, fan plot or spaghetti plot, with and without median lines).

We use eye-tracking techniques and exploration of the accuracy of decision tasks across expertise to address the following questions:

1. Does the presence of a median line and expertise affect gaze over the course of the decision-making process?
2. Does expertise affect gaze to the key over the course of the decision-making process?
3. Does expertise affect accuracy of decisions?

## 2. Methodology

### 2.1 Participants

Sixty-five participants took part in this study: twenty-two meteorology students, twenty-two psychology students and twenty-one graphic communication students recruited from the University of Reading (38 females, 27 males). Participants were aged 18–32 (M= 21.2) and had completed 0–4 (M=1.0) years of their respective degrees. Meteorology students are considered to have more training in graph reading, scientific data use, and quantitative problem solving as part of their degree and in qualifying for the course, than students on other degree courses which have less of a focus in these areas. Within this study, meteorology students were therefore considered to have greater expertise compared to the psychology and graphic communication students, although psychology students are also likely to have statistical knowledge and experience reading graphs. The research team involved academics who taught on each of these subjects and therefore can substantiate these generalisations.

### 2.2 Design and Procedure

Full methodological details are given in our companion paper, but to restate the core procedure: A hypothetical scenario of ice thickness forecast for a fictional location was

provided to participants. This type of forecast was chosen as is very unlikely to be one that is familiar to our participants to minimize any effects of preconceived notions of uncertainty. Participants were informed that they were making shipments across an icy strait and, using ice-thickness forecasts, had to decide whether to send a small ship or large ship. The small ship could crush 1-meter thick ice whereas the large ship crushes ice larger than this. There was a differential cost involved in this decision with small ship costing £1000 to send and the large ship £5000. They were additionally made aware that if the ice was thicker than 1-meter and small ship was sent, this would incur a cost penalty of £8000.

Ice thickness forecasts were presented in seven different types: deterministic line, box plot, fan plot and spaghetti plot. Each representation was presented with or without a median line. Each of these graph types was shown to represent 30%, 50%, and 70% probability of ice thickness exceeding 1 meter (See Fig. 1 for examples of each graph type). In this paper we only examined the decision-task question where participants were asked to select which ship (small or large) to send across an icy strait 72 hours ahead of time using a 72-hour forecast of ice thickness (see our companion paper Mulder et al. (2023) for further details on the hypothetical scenarios). While performing this task, participants wore an Eye link II eye-tracker headset which recorded eye movements of the right eye as they completed the survey. Head movements were restrained, and the eye tracker was calibrated to ensure accurate eye movement recording.

**2.3 Eye tracking apparatus**

Participants wore an EyeLink II (SR Research Ltd) eye tracker headset (See Fig 2 for pictures of the eye-tracker used with an example boxplot trial shown on the display; see https://www.sr-research.com/eyelink-ii/ for more details and pictures of the device) which recorded eye movements of the right eye at a rate of 500Hz as they completed the task. The EyeLink II is a high-resolution comfortable head-mounted video-based eye tracker with 0.5 deg average accuracy and 0.01 deg resolution that gives highly accurate spatial and temporal resolution. Participants gaze was precisely calibrated and re-calibrated throughout the study as necessary to maintain accurate recording. Each forecast, and task were presented on a 21-inch colour desktop PC with a monitor refresh rate of 75Hz. Participants were seated at a distance of 57 cm from the monitor and their head movements were minimized by a chin rest. Fixation location and its duration were extracted after study completion. Fixation was defined as times when the eyes were still and not in motion (i.e., no saccades were detected). These measures were used as proxies of the aspects of the forecasts were being attended to by participants as they made their decisions. These give a direct insight into the information and visual features that are salient when participants are

attempting to understand and use uncertainty in forecasting in order to make decisions. For
more information on methods used in eye-tracking studies, see Holmqvist et al. (2011).

**2.4 Data analysis**

Two interest areas were formed from a post hoc classification to address our research
questions (graph area and key). Three viewing periods across trials were created (early,
intermediate, late). The exact definition of early, intermediate, and late differed by type of
graph due to each style evoking slightly different viewing periods. Viewing periods for each
specific graph type were of equal bins divided across the average time to complete the
question and therefore ranged between 5 to 6 seconds. In this study, we report number of
fixations and total fixation duration.
In our companion paper (Mulder et al., 2023), our analysis of gaze was across all
experimental trials and all tasks. However, as we are concerned about the viewing period
and want to avoid effects of learning, we examine gaze when participants were faced with
each graph type for the first time. Repeated exposure to graph type and the demand to
make the same judgement may influence gaze patterns as informative parts of the figures
are located more swiftly. Therefore, six trials for each graph type for each participant were
examined. We analysed the accuracy of responses to this question (making the safe and
cost-effective choice of the two options) and gaze (number and total fixation duration).
Based on the results of our companion paper (Mulder et al., 2023), we further explore the
impact of the presence of a median line considering the viewing period, expertise and graph
type. We then focus on fixation towards the keys including viewing period, expertise, graph
type and the presence of a median line as variables. Data was analyzed using an Analysis of
Variance approach which tests for differences across the mean responses in cases where
there are multiple conditions or groups greater than two. Further post-hoc analyses
examining differences between specific pairs of conditions or groups were carried out using
t-tests which are Bonferroni corrected (this is a correction to the significance threshold
criteria to control for the number of comparisons carried out. See Baguley (2012) for
example). For both research questions a four-way mixed measures ANOVA was conducted
including graph type, presence of a median line and viewing period as within-subject
variables (i.e., all participants took part in all these conditions), and expertise as a between-
subjects variable (participants were grouped by expertise). Finally, we report the accuracy of
responses for the ice ship decision task highlighting any differences due to expertise.There
are a number of components to the output of the analysis of variance (ANOVA). Below we
provide a key which may help in understanding the output we report:
Key to Analysis of Variance (ANOVA) output
F: this is the inferential statistic test returned by the ANOVA which shows the proportion of variance
in the participant data explained by a model of the data that includes the levels of the independent
variable compared to that which can accounted for when that variable is not included (i.e., by
chance alone).
df: degrees of freedom are shown in brackets after the F value
MSE: Mean Square Error, this is the mean of variance accounted for by chance alone
p: shows the chances that the results would be found if there was actually no difference to be found.
The common threshold being 0.05 (5%). A p value less than 0.05 would be commonly labelled as
being significant, i.e., we were unlikely to have recorded the data we did if there was actually no
difference caused by the independent variable(s).
$\eta^2$: partial eta-sqaure. A measure of effect size. This gives an insight into the strength of the
effect of an independent variable. P values are affected by sample size where effect size
measures are not and so allow comparisons to eb made across variables.

**3. Results**

**3.1 Does the presence of a median line and expertise affect gaze over the course of**
**the decision-making process?**
Here, we examined how the presence of the median line influences eye movement
behaviour when considered across the viewing period from early to late stages, and different
levels of expertise, as well as the graph type. Table 1 shows a summary of the statistical
outcomes detailed in the paragraphs below, along with a short description of what they
show.
A main effect of presence of a median line was found for number of fixations and total
fixation duration made to the graph area, $F(1, 62)= 6.403$, *MSE*=32.747, *p*=0.014, $\eta^2$
=0.094; $F(1, 62)= 7.125$, *MSE*=2386741.96, *p*=0.01, $\eta^2$=0.103. More fixations were made,
and more time was spent fixating on the graph area of the display when no median line was
present (fixation count M=8.74; total duration M=2128.64) compared to when a median line
was provided (fixation count M=7.89; total duration M=1887.47).
A main effect of graph type was also found for number of fixations and total fixation duration
made to the graph area, $F(2, 124)= 15.098$, *MSE*=26.406, *p*<0.001, $\eta^2$=0.196; $F(2, 124)=$
16.810, *MSE*=1635280.256, *p*<0.001, $\eta^2$=0.213. Boxplots elicited more fixations, and more
time was spent fixating on boxplots (fixation count M=9.07; total duration M=2222.21) and
fan plots (fixation count M=8.71; total duration M=2091.04) compared to spaghetti plots
(fixation count M=7.17; total duration M=1710.92).
There was also a main effect of the viewing period for number of fixations and total fixation
duration made to the graph area, $F(2, 124)= 59.608$, *MSE*=36.762, *p*<0.001, $\eta^2$=0.488; $F(2,$
124)= 57.417, *MSE*=2294640.505, *p*<0.001, $\eta^2$=0.481. There was found to be a greater
number of fixations with longer dwell times on the graph area during early (fixation count
M=9.83; total duration M=2399.96) and intermediate (fixation count M=9.52; total duration
M=2284.11) viewing periods compared to later periods (fixation count M=5.60; total duration
M=1340.09).
There was no main effect of expertise on gaze behaviour measured by both fixation count
and total duration; $F(1, 62)= 0.536$, *MSE*=64.185, *p*=0.588, $\eta^2$=0.017; $F(1, 62)= 1.770$,
*MSE*=3970562.258, *p*=0.179, $\eta^2$=0.054, respectively.
As well as the main effects of median line, graph type and viewing period, there was an
interaction between the median line and viewing period for total fixation duration, $F(2, 124)=$
3.598, *MSE*=1543871.74, *p*=0.03, $\eta^2$=0.055. Less time was spent fixating the graph area
during the early and intermediate stages of viewing when a median line was present (Early
total duration M= 2174.97; Intermediate total duration M= 2137.79) compared to when no

median line was present (Early total duration M= 2624.96; Intermediate total duration M= 2430.43), $p<0.001$; $p=0.05$, respectively. However, no differences were found due to the presence (later total duration M= 1349.65) or absence (later total duration M= 1330.54) of a median line during the later stages, $p=0.896$. No other interactions were found to be significant. These findings support that the median line can reduce cognitive load; impacting the total fixation duration and number of fixations made on the graph area, particularly during early stages of the decision-making process, and adds to results from our companion paper that showed how fixation location was towards the median line when present, regardless of the type of graph.

|  | Number of Fixations | Total Fixation Duration | Summary |
|---|---|---|---|
| Main Effects |  |  |  |
| Median Line: Not Present vs Present | $F(1, 62)= 6.403$, $MSE= 32.747$, $p=0.014$, $\eta^2 =0.094$<br><br>Not present Mean (M) =8.74<br>Present M=7.89 | $F(1, 62)= 7.125$, $MSE= 2386741$, $p=0.01$, $\eta^2 =0.103$<br><br>Not Present M=2128.64<br>Present M=1887.47 | The presence of a median line on the graphs resulted in fewer fixations on the interest areas of the graph and key, with greater total fixation duration. |
| Graph Type: Boxplot vs Fan Plot vs Spaghetti Plot | $F(2, 124)= 15.098$, $MSE= 26.406$, $p<0.001$, $\eta^2 =0.196$<br><br>Boxplots Mean (M) =9.07<br>Fan plots M=8.71<br>Spaghetti plots M=7.17 | $F(2, 124)=16.810$, $MSE= 1635280$, $p<0.001$, $\eta^2 = 0.213$<br><br>Boxplots M=2222.21<br>Fan plots M=2091.04<br>Spaghetti plots M=1710.92 | Boxplots elicited more fixations and more time spent fixating the graph and key compared with fan plots and spaghetti plots |
| Viewing Period: Early vs Intermediate vs Late | $F(2, 124)= 59.608$, $MSE= 36.762$, $p<0.001$, $\eta^2 =0.488$<br><br>Early M=9.83<br>Intermediate M=9.52<br>Late M=5.60 | $F(2, 124)= 57.417$, $MSE= 2294640$, $p<0.001$, $\eta^2 = 0.481$<br><br>Early M=2399<br>Intermediate M=2284.11<br>Late M=1340.09 | Early viewing of plots shows a greater number of fixations on the graph and key with longer total fixation duration |
| Expertise: Meteorology vs Psychology vs Graphic communication | $F(1, 62)= 0.536$, $MSE= 64.185$, $p=0.588$, $\eta^2 =0.017$ | $F(1, 62)= 1.770$, $MSE= 3970562.258$, $p=0.179$, $\eta^2 =0.054$ | No significant differences found |
| Interactions |  |  |  |
| Median Line and Viewing Period | No significant interactions | $F(2, 124)= 3.598$, $MSE= 1543871.74$, $p=0.03$, $\eta^2 =0.055$ | Less time was spent fixating the graph area during the early and intermediate stages of |

| | | | |
|---|---|---|---|
| | | Early viewing period when median line was present M= 2174.97 vs not present M=2624.96, p<0.001 Intermediate, present M= 2137.79 vs not present M= 2430.43, *p*=0.05 Late, present M= 1349.65vs not present M= 1330.54, *p*=0.896 | viewing when a median line was present compared to when no median line was present No differences were found due to the presence or absence of a median line during the later stages |

Table 1. Shows a summary of the main significant statistical outcomes examining the effect of median line presence, graph type, viewing period and expertise on gaze behaviour as detailed in the text. All significant main effects and interactions are included along with important non-significant findings.

## 3.2 Is gaze to the key influenced by expertise and the viewing period during the decision-making process?

In order to examine how gaze parameters on the graph key change throughout the viewing period prior to the final decision, we extracted the number of fixations made to the key and their duration. Table 2 shows a summary of the statistical outcomes detailed in the paragraphs below, along with a short description of what they show.

A main effect of graph type was found for number of fixations and total fixation duration made to the key, $F(2, 124)= 42.900$, $MSE$=8.096, $p<0.001$, $\eta^2$=0.409; $F(2, 124)= 42.396$, $MSE$=574225.040, $p<0.001$, $\eta^2$=0.406. More fixations were made, and more time was spent fixating on fan plot keys (fixation count M=2.45; total duration M=626.79) compared to both boxplot (fixation count M=1.48; total duration M=387.75) and spaghetti plot keys (fixation count M=0.56; total duration M=127.13), and more fixations and time spent on boxplot compared to spaghetti plot keys.

There was a main effect of the viewing period on the number of fixations that were made to the key within the display, as well as the total amount of fixation, $F(2, 124)= 17.967$, $MSE$=6.593, $p<0.001$, $\eta^2$=0.225; $F(2, 124)= 21.003$, $MSE$=416719.669, $p<0.001$, $\eta^2$=0.253. More fixations and longer dwell time to the key occurred during the early (fixation count M=1.61; total duration M=407.15) and intermediate (fixation count M=1.99; total

duration M=515.33) viewing periods compared to later periods (fixation count M=0.90; total
duration M=219.20).
No main effect of the median line on gaze to the key, measured by both fixation count and
total duration, was found; $F(1, 62)= 0.175$, $MSE=7.574$, $p=0.677$, $\eta^2=0.003$; $F(1, 62)=$
$0.061$, $MSE=543399.152$, $p=0.805$, $\eta^2=0.001$, respectively. Nor was there a main effect of
expertise on fixation count and total fixation duration; $F(1, 62)= 0.251$, $MSE=10.191$,
$p=0.779$, $\eta^2=0.008$; $F(1, 62)= 0.141$, $MSE=730099.249$, $p=0.869$, $\eta^2=0.005$, respectively.
An €nteraction between the graph type and viewing period for fixation count and total fixation
duration was found, $F(4, 248) = 3.578$, $MSE=4.724$, $p=0.007$, $\eta^2=0.055$; $F(4, 248) = 4.260$,
$MSE=330504.612$, $p=0.002$, $\eta^2=0.064$., respectively. More fixations were made, and more
time was spent fixating the boxplot key during the early (fixation count M= 1.68; total
duration M=423.76) and intermediate (fixation count M= 2.06; total duration M=577.11)
stages of the viewing period compared to the later stage (fixation count M=0.71; total
duration M=162.39  $p<0.005$. Similarly, more fixations were made, and more time was spent
fixating the fan plot key during the early (fixation count M= 2.69; total duration M=695.64)
and intermediate stages (fixation count M= 3.10; total duration M= 791.37) compared to the
later stage (fixation count M=1.55; total duration M=393.37) $p<0.005$. However, no
differences were found between viewing periods for spaghetti plots, $p>0.05$. The reason for
less fixation being to spaghetti plot keys generally, and no differences overtime, could be
due to the intuitiveness of this form of plot and the simplicity of the key.

| Effect of… | Number of Fixations | Total Fixation Duration | Summary |
|---|---|---|---|
| Main Effects | | | |
| Median Line: Not Present vs Present | $F(1, 62)= 0.175$, $MSE=7.574$, $p=0.677$, $\eta^2=0.003$ | $F(1, 62)= 0.061$, $MSE=543399.152$, $p=0.805$, $\eta^2=0.001$ |  No significant differences found |
| Graph Type: Boxplot vs Fan Plot vs Spaghetti Plot | $F(2, 124)= 42.900$, $MSE=8.096$, $p<0.001$, $\eta^2=0.409$<br><br>Boxplots M=1.48<br><br>Fan plots M=2.45<br><br>Spaghetti plots M=0.56 | $F(2, 124)= 42.396$, $MSE= 574225.040$, $p<0.001$, $\eta^2=0.406$<br><br>Boxplots M=626.79<br><br>Fan plots M=387.75 | Fan plots elicited more fixations and more time spent fixating the graph and key compared with boxplots and spaghetti plots |

| | | Spaghetti plots M=127.13 | |
|---|---|---|---|
| Viewing Period: Early vs Intermediate vs Late | $F(2, 124)= 17.967$, $MSE=6.593$, $p<0.001$, $\eta^2=0.225$<br><br>Early M=1.61<br><br>Intermediate M=1.99<br><br>Late M=0.90 | $F(2, 124)= 21.003$, $MSE= 416719.669$, $p<0.001$, $\eta^2=0.253$<br><br>Early M=407.5<br><br>Intermediate M=515.33<br><br>Late M=219.20 | Early and intermediate viewing of plots shows a greater number of fixations on the graph and key with longer total fixation duration |
| Expertise: Meteorology vs Psychology vs Graphics | $F(1, 62)= 0.251$, $MSE=10.191$, $p=0.779$, $\eta^2=0.008$ | $F(1, 62)= 0.141$, $MSE= 730099.249$, $p=0.869$, $\eta^2=0.005$ | No significant differences found |
| Interactions | | | |
| Graph Type and Viewing Period | $F(4, 248) = 3.578$, $MSE=4.724$, $p=0.007$, $\eta^2=0.055$<br><br>Boxplot<br>Early M= 1.68<br>Intermediate M=2.06<br>Late M=0.71<br>p<0.0005<br><br>Fan plot<br>Early M= 2.69<br>Intermediate M=3.10<br>Late M=1.55<br>p<0.0005<br><br>Spaghetti plot<br>Early M= 0.45<br>Intermediate M=0.79<br>Late M=0.44<br>p>0.05 | $F(4, 248) = 4.260$, $MSE= 330504.612$, $p=0.002$, $\eta^2=0.064$<br><br>Boxplot<br>Early M=423.76<br>Intermediate M=577.11<br>Late M=162.39<br>p<0.0005<br><br>Fan plot<br>Early M=695.64<br>Intermediate M=791.37<br>Late M=393.37<br>p<0.0005<br><br>Spaghetti plot<br>Early M=102.05<br>Intermediate M=177.50<br>Late M=101.84<br>p>0.05 | Boxplots and Fan Plots show fewer fixations with less total fixation duration over viewing period but there was no effect of viewing period for spaghetti plots |

Table 2. Shows a summary of the main significant statistical outcomes examining the effect of median line presence, graph type, viewing period and expertise on gaze behaviour to the graph keys as detailed in the text. All significant main effects and interactions are included along with important non-significant findings.

## 3.3 Does expertise affect accuracy of decisions?

Mulder et al. (2020) found no significant difference in accuracy of decisions made between the graph types, just in the amount of uncertainty interpreted from them. Here, accuracy responses on the number of times participants correctly identified which ship would be most economically rational to send were measured considering expertise and probability of risk.


| | Meteorology | Psychology | Graphic Communication |
|---|---|---|---|
| 30% probability | 74% | 66.2% | 75.5% |
| 50% probability | 87% | 70.1% | 72.1% |
| 70% probability | 95.4% | 96.1% | 94.6% |

Table 3. presents accuracy results for all probabilities of risk for differing expertise. A small ship is the
correct ship to send for a 30% risk of ice thickness and a large ship for 50% and 70% risk levels.

Overall, participants were accurate in their choice of ship (Meteorology= 85.5%;
Psychology= 77.9%; Graphic communication = 80.7%); however, some differences were
apparent due to expertise. A one-way ANOVA shows differences in accuracy when
presented with 50% probability of risk, which is the most challenging task, $F(2,64)= 4.029$,
MSE=2.27, $p$=0.023, $\eta^2$=0.115. Multiple comparisons show meteorology students to be
significantly more accurate than psychology students in choosing the large ship during these
scenarios, $p$=0.035, and more accurate than graphic communication students, although this
difference is not significant, $p$=0.08. No differences between expertise were found for the
30% and 70% trials, $p$>0.05.

**4. Discussion and Conclusions**
As scientific information is increasingly being presented to non-specialists graphically, it is
important to consider how this information is delivered. This approach to open science, less
dependent on expert interpretation, is a natural development as general scientific literacy
increases and is welcomed by both scientific producers and consumers. As this approach
develops, it becomes much more important to have a clear understanding of the biases in
interpretation that results from different forms of data presentation. While relevant to many
fields of science, there is a particular need for this understanding in the environmental
sciences as environmental hazards increase and change.
Prior research presents mixed results, with some authors suggesting that when making
slight variations to graph representations that display uncertainty, decisions and
interpretations differ (Correll & Gleicher, 2014; Tak et al., 2015), whilst others show that
despite greater discrepancies in forecast representation, such as between graphic
visualisations and written forms, there are no differences (Nadav-Greenberg & Joslyn,
2009). Furthermore, few studies explore how experts and non-experts interpret forecast
information from different types of graphical forecast representations (Mulder et al., 2020).
The current research examines these areas further by using eye-movement techniques
considering expertise, and the viewing period during the decision-making process when
observing a range of graph types.
More economically rational responses to the ship decision were made by meteorology
students (greater level of expertise) during the most difficult scenarios. We found
participants, regardless of expertise, to spend less time fixating the overall graph when a
median line was presented, particularly during early and intermediate stages of viewing. This
provides more evidence for the anchoring bias suggested in previous papers (Mulder et al.,
2020). Participants focussed on the key for boxplots and fan plots more during early and
intermediate stages compared to later stages. This provides evidence that early stages of
viewing are more exploratory and towards informative areas (Buswell, 1935; Yarbus, 1967;
Antes, 1974; Nodine et al, 1993; Locher, 2006; Locher et al, 2007; Locher, 2015; Goldberg &
Helfman, 2010). However, considering the results and the differences found due to graph
type, spaghetti plots appear to be simpler to interpret, potentially reducing cognitive load
(Walter and Bex, 2021), corroborating the findings in Mulder et al. (2020) that the spaghetti
plot helped users interpret extreme values.
Overall, this study, together with the analysis in our companion paper (Mulder et al., 2023),
demonstrates that there are many challenges when presenting natural hazard data to both
experts and non-experts, the way that information is portrayed can impact interpretations
and decisions. It is important to note that the graph area and key discussed here are specific
to the particular tasks presented in this study and are used as indicators of the impact of
expertise, graph type and the viewing period. Furthermore, course of study within higher
education was used as a proxy for expertise, with meteorology students being regarded to
have higher levels. However, future research would benefit from examining behaviour and
decisions of academics and forecasters who would be considered as experts.
Responses to the ship decision (small or large) based on economic rationality supports the
importance of expertise as accuracy reduces dependent on the probability of ice thickness,
with those with greater expertise being more accurate during more uncertain situations.
While their accuracy was as low as others for 30% probability conditions, with a little less
uncertainty (50% probability of risk) accuracy improved more so than the other groups. This
suggests that they were able to use their expertise to understand the forecasts to inform
their decisions more effectively than the other groups. However, expertise appears to have
little impact on eye movement behaviour within our study. Differences between experts and
non-experts on decisions and interpretations of best-guess forecasts and their inference of
uncertainty have been reported previously (Mulder et al., 2020). However, Doyle et al.
(2014) found no differences in the use of probabilistic information for forecasts of volcanic
eruptions. Other contradictory evidence has also been reported testing numeracy as a
predictor for making economically rational decisions (Roulston and Kaplan, 2009; Tak et al.,
2015). Differences may be due to what "expert" means in these circumstances. As pointed
out, our sample used years of study as the expertise proxy and while showing some effect
may not reflect the decision-making and behaviour of those with many years of experience.
Thus, it may well be the case that those with greater expertise would show a more effective
use of forecast information provided both in terms of accuracy and more effective
information extract shown through eye movement differences not found in our sample.
The results show how median lines can reduce cognitive load drawing users to the central
estimate regardless of expertise. A median line reduces the perceived uncertainty in a
graphic, even when explicitly presented (Mulder et al. 2020), so use of a median line should
be used when the amount of uncertainty in the estimate is less critical to understand. Use of
the key within graphical representations can also impact interpretations of data. For forecast
providers this suggests that standard information design principles which seek to reduce
visual noise in data presentation and draw the user to the critical parts can have major
benefits for their ability to effectively communicate with both expert and non-expert end-
users.
More broadly, taken together the results reported here and those reported by Mulder et al
(2023) suggest that incorporating eye-tracking and other techniques from cognitive science
into the process of the design of forecast communication tools could be extremely fruitful.
These techniques are now well-established with technology that makes them relatively
cheap to set up and use. Graphical presentation of geo-scientific forecasts can happen with
a range of breadth and longevity of communication in mind. While eye-tracking and related
techniques would not be appropriate for all purposes, where graphics are being developed
for routine and wide use, for example routine weather forecasts, this kind of approach would
be a very valuable addition to end-user engagement. One obvious extension to the work in
the two parts of this study is applying the same techniques to well-known and widely used
geo-scientific forecast graphics.

**5. Author contributions**
Louis Williams: Conceptualization, Investigation, Formal analysis, Writing – original draft
preparation
Kelsey Mulder: Writing – review & editing
Andrew Charlton-Perez: Funding acquisition, Writing – review & editing
Matthew Lickiss: Writing – review & editing
Alison Black: Funding acquisition, Writing – review & editing
Rachel McCloy: Funding acquisition, Writing – review & editing
Eugene McSorley: Conceptualization, Resources, Writing – review & editing
Joe Young: Funding acquisition
*Acknowledgments.* We thank our eye-tracking study participants. This research is
funded  by the Natural Environment Research Council (NERC) under the Probability,
Uncertainty and Risk in the Environment (PURE) Programme (NE/J017221/1). Data created
during the research reported in this article are openly available from the University of
Reading Research Data Archive at http://dx.doi.org/10.17864/1947.110

The authors declare that they have no conflict of interest.

### Ethical Statement

The University of Reading Ethics Board approved the study, and the study was conducted in
accordance with the standards described in the 1964 Declaration of Helsinki. Participants
provided written informed consent. The authors declare that there is no conflict of interest.

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

Figure Legends
Figure 1. The four forecast representations used in this analysis: (a) deterministic (using only
the median line), (b) and (c) spaghetti plot, (d) and€) fan plot, and (f) and (g) box plot.
Uncertainty forecasts were shown both with median lines (b,d,f) and without median lines
(c,e,g). All forecasts represent the same information: three of 10 model runs show ice
greater than 1-meter thick. The same plots were produced for 50% and 70% chance of ice
greater than 1-meter thick (not shown). The dotted line in each graphic shows 1-meter ice
thickness, the threshold the participants predicted.
Figure 2. On the left are pictures of the head-mounted eye-tracker, EyeLink II (SR Research
Ltd), used to record participant's eye movements while taking part in the study with an
example of boxplot trial shown on the display. Note that the small diagonal line visible on the
top right of the display screen (bottom left photo) is an artefact of the photograph and the
refresh rate of the monitor. On the right, composite heat maps are shown. These show the
accumulation of the duration of eye fixations (in milliseconds) of all participants for the ship
decision (a,b) and maximum ice thickness (c,d) tasks. Heat maps are shown only for the
spaghetti plot with (a,c) and without (b,d) median lines. Heat maps for the other forecast
representations can be found in the Appendix B of Mulder et al (2023). Between each
question, there was a cross present to help participants focus back to to the centre of the
screen prior to moving on. Artefacts of this centering can be seen on the heat maps.