# Peer review of "Understanding representations of uncertainty, an eye-tracking study part II: The effect"

_EGUsphere, 2022_

## Author Comment (AC1)

Reviewer 1
This paper is linked/an extension to a previous study, but it does have a novel aspect and there is a clear rationale for the study.

I feel there needs to be more discussion examining the decisions linked to expertise. I feel the findings need to be slightly toned down to reflect the lack of significant differences in decision making capacity relative to expertise as there was a difference at 50% but not 30 or 70%.

Response: Further discussion of the findings and links with expertise have been added to the discussion. Furthermore, the findings have been more explicit at this point and so toned down.

It may be minor but introducing someone else's results (line 238, 279) in your results section is quite confusing.

Response: This has been amended. Both this and a similar justification has been moved into the introduction to bolster the motivation for our research questions. There is now no direct report of results from previous papers made in this paper.

Also relative to the experts and non-experts, I would agree with the comments made in line 336 considering the recruitment of true experts in the area as a comparison. It would also help to link back to why expertise had little impact on eye movement behaviour (line 341)

Response: This has been added.

---

## Author Comment (AC2)

Reviewer 2

I enjoyed reading the manuscript, however there are some sections of the paper that need strenghtening.

As a first comment, I would like to see this paper a stand alone and most of the times the authors make reference to another paper (which is not yet published).

Response: The linked paper has now been accepted into this Journal and we would expect that this paper would be obviously linked to it via journal website. They would be more obviously companion papers than has been made obvious during the review process. We would much prefer to keep this arrangement and not have separate the two. They are both of a piece and length was the main motivating factor for separating them.

My specific comments are as follows:

1. L48 Not clear how Covid19 further impacted communication of uncertainty.

Response: We were alluding to the sharp increase in graphical communication about communication that occurred at that time. During the Covid-19 pandemic the public were presented with a great deal of uncertain scientific data in various graphical forms, many of which they were unfamiliar with. These were used by governments to explain why decisions have been reached and why actions are being asked of them (eg stay 2 metres away from others, wash hands). From this perspective, it is important to understand how best to communicate uncertainty to the general public and how best to graphically support them to reach an appropriate understanding. We have added text to the first paragraph to make this link more explicit.

2. L56 The problem is not well articulated, and the readers would benefit from an extensive literature search of how end-users have struggled with communication of uncertainty. A lot of work has been done in the past and the authors can make reference to how some the challenges to communicating uncertainty (eg. Spiegelhalter, D., Pearson, M., and Short, I.: Visualizing Uncertainty About the Future, Science, 333, 1393–1400, https://doi.org/10.1126/science.1191181, 2011).

Response: A more extensive literature review of the communication of uncertainty has now been included at the start of the introduction.

3. L78-- L85 A lot has been done on this subject of using visuals for communicating uncertainty and this would be important to emphasize this in this section (see Milne, A. E., Glendining, M. J., Lark, R. M., Perryman, S. A. M., Gordon, T., and Whitmore, A. P.: Communicating the uncertainty in estimated greenhouse gas emissions from agriculture, J. Environ. Manage., 160, 139–153, https://doi.org/10.1016/j.jenvman.2015.05.034, 2015; Zikmund-Fisher, B. J., Fagerlin, A., and Ubel, P. A.: Improving Understanding of Adjuvant Therapy Options by Using Simpler Risk Graphics, Cancer, 113, 3382–3390, https://doi.org/10.1002/cncr.23959, 2008)

Response: This has been folded in with our response to point 2 by changing the introduction to be a more extensive literature review of the communication of uncertainty.

4. L112-- L122 I don't think this is very necessary to discuss the relation between the two papers at length, I would focus on the problem this manuscript is focusing on and why it is important. The authors introduce the concept of eye-movement but quickly divert to another paper. Focus should remain on this paper. Again I would expand on the complex methods the authors mention in passing and why they are relevant. A supplement of the test methods would be useful.

Response: The brief digression has been removed. Methods have been expanded as requested.

5. L125 i dont see how these research questions analyse/capture the differences in the end-user groups

Response: We confess to finding this comment difficult to understand. The research questions each mention that the effect of expertise on gaze control will be examined. Each one of these directly addresses differences in the expertise of different end-user groups.

6. L147 Here again the authors mention the companion paper (which cant be found anywhere)- A summary of the methodolgy would be useful.

Response: We can only apologize about this – our intention was for the two papers to be reviewed together but this has not possible. The companion paper has now been accepted for publication in this journal the web links between the two can be made more explicit and cross checking by readers will be much easier. However, in response to this comment we have added further methodological information so that those details regarding the study reported in this paper are self-contained.

7. L153 a graphical illustration would be more useful here and an explanation why the Eye link II eye tracker is useful or important on L156 is needed.

Response: Further details about the eye tracker have been given as has a direct link to its manufacturers specification page which also has pictures of the head mounted system.

8. I agree with the first referee, why would you report results from another paper, this paper should stand alone and casually make reference to the other paper when needed.

Response: As stated in our response to Reviewer 1, this has been amended. This has been moved into the penultimate paragraph of the introduction as justification for motivating our research questions. There is no direct report of results from previous papers made in this paper.

9. L193 to 198 is the justification and it seems misplaced.

Response: As per the previous response, this has now been placed in the penultimate paragraph of the introduction.

10. The information presented from L199 to 204 should be presented in a table. Similar comment for L211 to 223.

Response: This would require multiple tables; 3 for each section. We feel that including these would not help and would be confusing for the reader. We did try this approach, but the tables needed seems excessive when the results are well captured and are clearer by placing them directly in the text.

11. L238 seems misplaced as well as it belongs to discussion

Response: As per the response to point 8 and 9, this has now been placed in the penultimate paragraph of the introduction.

12. L259 to 266 should be presented in a table.

Response: see response to point 10. This suggestion introduces the same issues with clarity of presenting the findings.

13. L290 A plot of the accuracies for the different clusters would communicate the research findings here!

Response: The data are presented in tabular form rather than a figure. This has the benefit of showing the data clearly as numbers. Replacing this with a figure would certainly show the same data but not give the readers explicit access to the numbers. We are keen to convey this information and reticent about readers having to estimate the actual numbers. One option would be to have the figure and then place numbers in the text but this would seem to introduce a further step into the results section that is not needed.

---

## Author Response (AR2)

Reviewer 1:

1. The additional information added to the introduction does help to provide a clearer rationale and novel/standalone aspect to this paper. Although, the paper discusses scientific information with uncertainty in the introduction (highlighting COVID_19) and then also includes more specific information on geoscience and natural hazard data in the introduction/discussion. Does the context matter and if so how is the information on COVID_19 relevant and what does that mean for expertise and audience who are interpreting the data? Arguably the cohort in this sample have a degree of familiarity with interpreting data (albeit not in the context) as students in Higher Education and this may not cover the spectrum of ability represented in the introduction and discussion as the audience to the COVID_19 and non-specialists.

In line 74 the authors discuss the "desired outcomes of the communicator and the needs and abilities of the audience" and I think this information needs to be considered in the context of the participants and the figures presented in this study.

**Response: The point being made at the beginning of the introduction is that scientific information can be presented in several ways and the nature of the representation may be more or less familiar to an audience. COVID-19 was simply used a general example of this issue. It is mentioned once in the second sentence of the introduction and then not mentioned again afterwards. There is no attempt in the paper to state that COVID-19 is relevant beyond this.**

**However, there are generalities that can be drawn from previous research in scientific communication: context clearly matters, in terms of the information being communicated, and the audience being communicated to. As the reviewer mentions, these are all points raised in the introduction and discussion. We are careful throughout to state who our participants were and, given their areas of study, the differing levels of expertise that they clearly have when dealing with the materials we employ in the research reported. So, in this sense the findings we report are context dependent. However, we are careful to highlight what conclusions can be drawn from our findings that would apply to other contexts in the discussion. The key one being that anchoring lines draw the eyes and therefore attention and act to reduce cognitive load in decision formation.**

2. There are also some points of clarification in the results and discussion
• Results
o For example, in line 344 and 345 in section 3.2 - the wording (In order to examine fixation to the key over different periods of the decision-making process") could be clarified and explained especially in relation to the sub heading of 3.2 title

**Response: This has been clarified to read:**
**"In order to examine how gaze parameters on the graph key change throughout the viewing period prior to the final decision, we extracted the number of fixations made to the key and their duration."**

3. Also in 3.3 the results section starts by explaining the results of the companion paper.

**Response: This is not an explanation of the companion paper, this is reference to Mulder et al 2020, an earlier paper – Not Mulder et al (2023).**

4. Discussion
o In Line 417- 418 "More economically rational responses to the ship decision were made by meteorology students (greater level of expertise) during the most difficult scenarios"- a greater explanation for this finding is required, especially as eye movements did not change. In line 444 the authors refer to meteorology students "use their expertise". What expertise are they using and can this expertise be transferable or what else needs to communicated to non-specialised audience to help with interpretation if expertise is context specific?

**Response: Metrology students made more economically rational decision as shown by the final section of the results (3.3 Does expertise affect accuracy of decisions?): Metrology students were more accurate in their choice of ship especially so when the task is a challenging one (50% probability of risk). Given the lack of differences found in gaze responses this suggests that information extracted from the graphs as a function of expertise differs in terms of how it is used by participants to inform their final decisions. We state this conclusion and discuss it on lines 441 onwards:**

**"Responses to the ship decision (small or large) based on economic rationality supports the importance of expertise as accuracy reduces dependent on the probability of ice thickness, with those with greater expertise being more accurate during more uncertain situations. While their accuracy was as low as others for 30% probability conditions, with a little less uncertainty (50% probability of risk) accuracy improved more so than the other groups. This suggests that they were able to use their expertise to understand the forecasts to inform their decisions more effectively than the other groups. However, expertise appears to have little impact on eye movement behaviour within our study….."**

5. Also the terms companion paper and the reference is used interchangeably throughout despite the text saying from X point it would be referred to as companion paper.

**Response:**
**The way the companion paper was referred to in the paper depended on the context of the text surrounding it. However, we have amended the manuscript to remove any mention of the companion paper as a reference alone.**

Reviewer 2:
1. Please add a graphic of the eye tracker II in the paper or supplement. The provided link is not working.

**Response:**
**We have checked and the link works fine. A direct click in the word document or a copy and paste from there takes us straight to the Eyelink II description page on the SR Research website. We are not clear why it has not worked in this case. We are happy to add an image if it would help and would value advice on this from the editor and the journal regarding the desirability of this.**

---

## Author Response (AR3)

Thank you very much for your helpful comments on the paper. We have made all of the edits asked for.  We would hope that these changes have increase the readability of the paper.

Editor Comments:

1. Your manuscript can benefit significantly from adding at least two figures (method section) and an additional table (result section). This was also pointed out by the reviewers. Please add a figure showing ice thickness forecasts in seven different types. This should help readers visualize the different graph types you use with your research participants. The second figure that would be useful is a photo (or cartoon drawing) of the eye tracking device together with a screenshot of what participants look at and how the eye tracking appears on the screen. Please note that including a link to the device website does not suffice as links can change or be removed over time. It also keeps the reader on the manuscript page instead of sending them to another page.

**Response:**
**Added a figure showing graph type examples as Figure 1.**
**Added a figure (Figure 2) showing pictures of the eye-tracker from two angles including one in which a trial is shown on the display used in the experiment. We have also included a map showing the concentration of fixations on four example trials, known as heat maps. It should be noted that no eye-tracking is shown to the participants as they examine the graphs. They do not see an indication on the display showing them where their gaze lies.**

2. Sections 3.1 - 3.3 (results) are difficult to follow. Please list all the stats in a table that you can refer to in the text. This was also mentioned by the reviewers. Consider including an image of each graph in this table to help the readers identify quickly what data go with what graph type/area, and provide a caption that explains all the terminologies, and abbreviations you use in the table and text (e.g., F, MSE, etc.). Not all GC readers are familiar with these terms.

**Response: To ease this difficulty we have added two tables (Table 1 and 2) summarizing the statistical outcomes for sections 3.1 and 3.2 to the results section and pointed to them in the text in the first paragraph of each section. We would hope the addition of Figure 1 showing graph types will help readers successfully link up the results with each graph type. We have also provided a key to the analysis of variance outcomes at the end of the method section in the Data Analysis subsection.**

3. Please explain your stat work (mixed measure ANOVA - four-way vs. one-way) in the methods section and explain why this approach was chosen. Not every reader is familiar with this method. Also, consider moving line 294-296 to the methods section.

**Response: we have moved text from the beginning of the results section into the data analysis section (as suggested) and added to this with more detail about the ANOVA design and why this approach was adopted. We have also added a reference to this should the reader want to explore the analysis of variance any further.**

4. You have measured fixation frequency and duration. How about eye blinking and other types of eye movements that may be important to consider in the interpretation of your

results. Are these measurable, and if so, will they be relevant to the conclusions of this study?

**Response: The eye-tracking we carry out does return many other measures such as eye blinks, pupil size, saccade direction, its duration and velocity among many others. Some of these would be illuminating when it comes to examining how people look at graphical information over time in order to support their decision making. However, the measures we report are general measures that give a very good overview of gaze behaviour and information seeking when faced with these types of graphs. Other measures would be interesting to examine in other experiments, for example, when the physical layout of each graph type was directly manipulated in an organised way, such as manipulating the location of the key, but that was not the case here.**

**Eye blinks themselves might not necessarily be interesting here as they are commonly taken as a measure of anxiety and emotional reaction, which was not the focus of this study, but they may possibly reflect task difficulty which could be of interest when faced with unusual graphs and required to make decisions. Thank you very much for the suggestion we will careful consider this for our future work.**

---

## Author Response (AR4)

Thank you very much for your continued helpful suggestions for corrections and clarification. They are very much appreciated. We have addressed them as outlined below.

Best wishes
Eugene

Editor's Comments:

1. Line 230: Please consider deleting the first sentence in section 2.2 and reference the paper in the text: Full methodological details are given in our companion paper, but to restate the core procedure.

**Response: We have deleted the sentence and added the reference.**

2. Lines 252-254: Please revise the sentence to "Participants wore an EyeLink II (SR Research Ltd) eye tracker headset (Fig 2) which recorded eye movements of the right eye at a rate of 500Hz as they completed the task" and delete the link to the product.

**Response: This sentence has been revised as suggested.**

3. Line 257: Please explain what "0.5 deg average accuracy and 0.01 deg resolution" mean in the context of your study.

**Response: The following information has been added to the sentence in parentheses: "The EyeLink II is a high-resolution comfortable head-mounted video-based eye tracker with 0.5 deg average accuracy (offset between actual gaze location and that recorded) and 0.01 deg resolution (dispersal of gaze locations during fixations) that gives highly accurate spatial and temporal resolution."**

4. Line 262: Please insert "Fig 2" at the end of sentence "[...] by a chin rest."

**Response: this has been added**

5. Line 291: Please insert "(also known as ANOVA)" after Analysis of Variance.

**Response: This has been added**

6. Lines 301-316: Please move these lines to the caption for Table 1, and correct typos in lines 315-316.

**Response: The key to the ANOVA output has been moved and typos have been corrected**

7. Line 314: Typo (Eta-squared).

**Response: this has been corrected**

8. Line 328 onward: Not sure how useful it is to report all the stats in the text. It can be

distracting. I wonder if referring to Table 1 suffices (also see comment below about Table 1).

Table 1: Thanks for putting this together. However, it is still not quite easy to read, understand, and compare data for different graph types. Adding more columns may fix this issue. For example, consider creating a separate column for each of these values: F, df, MSE, M, and Eta-squared.

Table 2: Same comment as above.

**Response: We have changed the text in the main body of the paper to remove all mention of statistical output other than to give an indication of the p-value. Reference is made to the tables. The tables have been changed to show the statistical outcomes only with each element of the ANOVA output separated into columns. We hope this improves readability.**

9. Line 473: Typo (demonstrate)

**Response: this has been corrected**

10. Line 475: Please delete 'discussed here'

**Response: this has been deleted**

11. Line 481: Typo (support) - Also lines 481-483 are complex and vague, please consider revision.

**Response: Typo has been corrected and the sentence has been amended to aid clarity: "Responses to the ship decision (small or large) based on economic rationality support the importance of expertise. While accuracy generally reduces dependent on the probability of ice thickness, those with greater expertise are less prone to this and are more accurate during more uncertain situations."**

12. References: Please add a "period" after "al." for all references cited in your manuscript. Example: (Locher et al., 2007) and for those with two authors, replace '&' with 'and'.

**Response: These have been corrected for all references**

13. Figure 1 and 2: Great additions. Thanks for putting these together. In Fig 2 caption, you state "Artefacts of this centering can be seen on the heat maps." Could you please add a label to the figure to indicate this? I am not sure if it can be seen by everyone. I also suggest deleting "Note that the small diagonal line visible on the top right of the display screen (bottom left photo) is an artefact of the photograph and the refresh rate of the monitor." It might be more confusing to mention it. It may be safe to assume that most people won't even recognize this. Just a suggestion.

**Response: We have amended the text in the figure legend to help clarify this**
**"Please note that between each question, there was a cross present to help participants focus back to the centre of the screen prior to moving on to the next trial. This central**

start position resulted in collections of fixations in the centre of the displays and can be seen on all of the four heat maps shown. It is most clear on the top right heat map."